# The Microenvironment of the Pathogenesis of Cardiac Hypertrophy

**DOI:** 10.3390/cells12131780

**Published:** 2023-07-04

**Authors:** Farhad Bazgir, Julia Nau, Saeideh Nakhaei-Rad, Ehsan Amin, Matthew J. Wolf, Jeffry J. Saucerman, Kristina Lorenz, Mohammad Reza Ahmadian

**Affiliations:** 1Institute of Biochemistry and Molecular Biology II, Medical Faculty and University Hospital Düsseldorf, Heinrich Heine University Düsseldorf, 40225 Düsseldorf, Germany; farhad.bazgir@hhu.de (F.B.); j.nau@hhu.de (J.N.); 2Stem Cell Biology, and Regenerative Medicine Research Group, Institute of Biotechnology, Ferdowsi University of Mashhad, Mashhad 91779-48974, Iran; s.nakhaeirad@ferdowsi.um.ac.ir; 3Institute of Neural and Sensory Physiology, Medical Faculty and University Hospital Düsseldorf, Heinrich Heine University Düsseldorf, 40225 Düsseldorf, Germany; ehsan.amin@hhu.de; 4Department of Medicine and Robert M. Berne Cardiovascular Research Center, University of Virginia, Charlottesville, VA 22908, USA; mjw5mc@hscmail.mcc.virginia.edu; 5Department of Biomedical Engineering, University of Virginia, Charlottesville, VA 22908, USA; jsaucerman@virginia.edu; 6Institute of Pharmacology and Toxicology, University of Würzburg, Leibniz Institute for Analytical Sciences, 97078 Würzburg, Germany; lorenz@toxi.uni-wuerzburg.de

**Keywords:** cardiac hypertrophy, cardiomyocytes, heart failure, myocardial microenvironment, myofibroblasts, pressure overload, scar formation, vasoactive hormones

## Abstract

Pathological cardiac hypertrophy is a key risk factor for the development of heart failure and predisposes individuals to cardiac arrhythmia and sudden death. While physiological cardiac hypertrophy is adaptive, hypertrophy resulting from conditions comprising hypertension, aortic stenosis, or genetic mutations, such as hypertrophic cardiomyopathy, is maladaptive. Here, we highlight the essential role and reciprocal interactions involving both cardiomyocytes and non-myocardial cells in response to pathological conditions. Prolonged cardiovascular stress causes cardiomyocytes and non-myocardial cells to enter an activated state releasing numerous pro-hypertrophic, pro-fibrotic, and pro-inflammatory mediators such as vasoactive hormones, growth factors, and cytokines, i.e., commencing signaling events that collectively cause cardiac hypertrophy. Fibrotic remodeling is mediated by cardiac fibroblasts as the central players, but also endothelial cells and resident and infiltrating immune cells enhance these processes. Many of these hypertrophic mediators are now being integrated into computational models that provide system-level insights and will help to translate our knowledge into new pharmacological targets. This perspective article summarizes the last decades’ advances in cardiac hypertrophy research and discusses the herein-involved complex myocardial microenvironment and signaling components.

## 1. General Introduction

Myocardial remodeling associated with cardiac hypertrophy is one of the critical causes in the development of heart failure. The pathogenesis of heart dysfunction is one of the primary causes of morbidity and mortality in elderly people [1].

Cardiac hypertrophy is the most frequently compensatory or adaptive process to numerous physiological or pathological conditions (Table 1) [2]. Hypertrophic enlargement is characterized by an increase in the cell size of cardiomyocytes. The heart can dynamically change its muscle mass to cope with the stimuli of development, physiological conditions of exercise and pregnancy, or pathological disease stimuli (Table 1) [3]. Increased workload as a consequence of volume or pressure overload due to pathological or physiological stimuli increases tension on the cardiac walls of the heart chambers [4,5]. This ultimately triggers stress signals released by different cell types of the microenvironment to compensate for the wall tension increase, resulting in a hypertrophic growth response [4,5]. Individual cardiomyocytes can increase in length and/or width in response to hypertrophic stimuli depending on the intracellular signaling cascades involved [6,7,8].

Hypertrophy, or the enlargement of individual muscle fibers, is the primary mechanism by which skeletal muscle mass increases during postnatal development. A similar process may be induced in adult skeletal muscle in response to contractile activity, satellite cell proliferation and fusion, which increases the number of myonuclei. This event may also play a role in muscle growth during early but not late stages of postnatal development and in some forms of muscle hypertrophy [9]. Likewise, the number of endothelium and mesenchymal cells can also increase from birth through early adulthood, but on the other hand the entire complement of cardiomyocytes is created during pregnancy and remains nearly constant throughout the human lifespan. Early infancy has the highest levels of cardiomyocyte exchange, which steadily declines during life to 1% per year in maturity in processes that increase very modestly in the vicinity of cardiac injury [10,11]. Important to note that cell duplication is not always a result of cardiomyocyte cell cycle activity. In contrast, multinucleation and polyploidization occur during various phases of development including heart development as a result of premature cell cycle exit [12].

Physiological and pathological cardiac hypertrophy are associated with distinct molecular characteristics (Table 1) involving alterations in the expression of fetal genes, and contractile and calcium-handling proteins [13]. A major molecular characteristic of pathological hypertrophy is the re-expression of fetal genes. Pathological settings such as hypertension cause the induction of the stress program that involves increased expression of atrial natriuretic peptide (ANP), B-type natriuretic peptide (BNP), and alpha-skeletal actin (α-sk actin) [14]. In contrast, an important characteristic of physiological hypertrophy is the absence of molecular stress programs [15]. In addition, expression of cardiac contractile proteins, such as alpha- and beta-myosin heavy chain and calcium-handling proteins, e.g., sarcoplasmic reticulum Ca^2+^-ATPase 2a (SERCA2a) remain unchanged during physiological cardiac hypertrophy, whereas pathological cardiac hypertrophy is closely associated with alterations in the above-named genes and proteins [14].
cells-12-01780-t001_Table 1Table 1Characteristics of physiological and pathological cardiac hypertrophy ^1^.CharacteristicPhysiological Cardiac HypertrophicPathological Cardiac HypertrophicStimuliexercise, pregnancyi.a. pressure or volume overload Cardiomyocyte sizeincreasedincreasedConcentric or eccentriceccentric > concentricconcentric or eccentricAdaptivityyesinitially yes/advanced maladaptiveContractilitypreserved or increasedpreserved or decreased**Cardiac metabolism**

Fatty acid oxidationincreaseddecreasedGlycolysisincreasedincreased**Structural and functional**

ReplacementnoyesInterstitial fibrosisnoyesCardiomyocyte apoptosisnoyesCapillary networksufficientinsufficient**Molecular characteristics**

Fetal gene expressionunmodifiedupregulatedContractile linked genesInflammationnormal or increasedunmodifieddownregulatedincreasedCardiac functionnormal or increaseddepressedReversibleyesnoHeart failureunlikelyprone^1^ Adapted from Bernardo et al., 2010 [15] and, Nakamura and Sadoshima, 2018 [14].


While the events associated with physiological hypertrophy are generally reversible, those associated with pathological cardiac hypertrophy are commonly irreversible and impose a high risk of heart failure (Table 1). A common disease stimulus, such as long-standing hypertension usually causes pressure overload and increases systolic wall stress [16]. In this case, individual cardiomyocytes typically grow in width more than in length, leading to the thickening of the cardiac walls, a condition referred to as concentric hypertrophy [3,17]. Hypertrophic changes have been rationalized employing Laplace’s law, which says wall stress (or tension) is an inverse function of wall thickness (tension = [pressure × radius]/2 × wall thickness). Thus, compensated growth of the cardiac muscle is a physiological response to decrease wall tension and thereby maintain cardiac pump function [4]. Prolonged pathological stress, however, causes maladaptive changes at the cellular and molecular level resulting in pathological cardiac hypertrophy. Untreated pathological cardiac hypertrophy predisposes individuals to heart failure, arrhythmia, and sudden death [6,7].

Triggers of pathological cardiac hypertrophy include extrinsic drives such as pressure overload due to long-standing hypertension or valvular stenosis, as well as volume overload due to mitral regurgitation or aortic insufficiency (Table 1), loss of contractile mass (myocardial infarction), or intrinsic causes such as hereditary defects [2,18]. Although a notable feature of physiological and pathological cardiac hypertrophy is the increase in heart size, pathological cardiac hypertrophy involves the loss of myocytes and fibrotic replacement, leading to cardiac dysfunction, heart failure, and/or sudden death [19,20]. Despite views that the length of stress has a significant impact on the distinction between pathological and physiological cardiac hypertrophy, the nature of stress and the intracellular signaling cascades involved are thought to be more important in the development of maladaptive cardiac dysfunction than the chronic duration of exposure [21].

The role of the cardiac microenvironment and intercell communication in the cardiac niche is becoming increasinglyevident for future studies and therapeutic interventions, as also recently highlighted by Tazhor et al. [22], challenging the traditional view of the heart as a cardiomyocyte-centric and targeted regenerative and therapeutic target in cardiovascular disease. Future therapeutic and mechanistic investigations should take into account the fact that immune cells, fibroblasts, and endothelial cells collectively outnumber cardiomyocytes by a significant margin as the resident cells in the heart, making this viewpoint increasingly important as a crucial element in the study of the intercellular communications and the treatment of heart disease. Therefore, the aim of this review article was to focus on these processes related to the onset, progression, and pathogenesis of hypertrophic cardiomyopathy and to complement previous work by incorporating molecular axes and details of intercellular communication in the cardiac microenvironment that have not yet been illuminated.

## 2. An Interplay of Different Cells in Hypertrophic Remodeling

The heart consists of various cell types, including myocytes, endothelial cells, fibroblasts, vascular smooth muscle cells, sympathetic neurons, and immune cells, which collectively account for a synchronized cardiac function [23,24]. However, it has been shown that owing to their enormous size, cardiomyocytes in particular account for the majority of heart mass, increase in size and reprogram transcription in the process of cardiac hypertrophy [2,25]. Communications between cardiomyocytes and non-myocytes lead to the secretion of bioactive mediators, which operate in an autocrine and paracrine manner (Table 2). This is followed by microenvironmental stimulation of different cell types and the activation of various signaling pathways within the cells (Figure 1 and Figure 2) [26,27]. Altogether these complex processes result in cardiomyocyte hypertrophy, fibroblast hyperplasia, interstitial tissue composition changes, and remodeling of the ventricular chambers [28].

### 2.1. Fibroblast Remodeling

Pressure overload triggers resident cardiac fibroblasts originating from the epicardium and endocardium to undergo rapid expansion and activation, rather than previously reported hematopoietic precursor-derived fibroblasts or endothelial-to-mesenchymal transition (EndMT) as a contributing source (Figure 1 and Figure 3) [29,30]. Despite this, the exact origins of cardiac fibroblasts as well as the delineation of their characteristics and plasticity remain a field of current investigation and controversy [31]. Like cardiomyocytes, fibroblasts respond to external stress stimuli, but in a slightly different manner. Mechanical stress promotes fibroblast differentiation to a myofibroblast-like phenotype (Figure 1 and Figure 3) [32,33], which has been shown to develop from tissue-derived fibroblasts rather than endothelial or smooth muscle cells [30]. Myofibroblasts overproduce and release extracellular matrix (ECM) components and pro-hypertrophic mediators, including Transforming growth factor-beta (TGF-ß) (Table 2), and are engaged in a wide range of pathological conditions, particularly fibrosis and tissue remodeling [34]. Enhanced release of ECM by myofibroblasts contributing to mechanical stiffness accompanied by increasing fibrosis evolves into severe consequences causing cardiac diastolic dysfunction [35]. Moreover, progressing fibrosis can affect systolic function by building a barrier between the resident cardiomyocytes, thereby provoking defective electrical coupling within the myocardium [36]. Additionally, an increased level of ECM, such as collagen, can disrupt the oxygen diffusion capacity leading to hypoxia in the affected myocytes a process that may further enhance pathological remodeling [37]. In conclusion, cardiac fibroblasts react to pressure overload-induced injury with activation, accumulation, and excessive ECM deposition (Figure 1 and Figure 3). The resulting conditions including mechanical stiffness, myocyte uncoupling, and ischemia comprise key contributors to heart failure [29]. These lines of evidence also emphasize the identification of mechanical stress in cardiac hypertrophy as an independent risk factor for arrhythmias, myocardial infarction, and sudden death [19].

### 2.2. Endothelial Cell Activation

In response to pressure overload, cardiac endothelial cells, similar to cardiac fibroblasts, are capable of changing their phenotype (Figure 1). It has been reported that endothelial cells can undergo an EndMT, differentiate into myofibroblast-like cells, and thereby contribute to cardiac fibrosis [38]. Others outlined that EndMT recruits circulating hematopoietic progenitors to the heart thereby generating significant numbers of cardiac fibroblasts (reviewed in [39]) but also their origin from tissue-resident fibroblasts is being discussed [29,30]. Altogether, left ventricular myocardial tissue of end-stage cardiac failure patients revealed dramatically increased expression levels of EndMT-related genes [40], indicating the need for further investigation to clarify the exact contribution of EndMT.
Figure 1**A microenvironmental model of pressure overload-induced cardiac hypertrophy.** The model also illustrates multiple cell types’ substantial roles and reciprocal interactions in the myocardium. In response to pressure overload cardiomyocyte and non-myocardial cells are transformed into an “activated state”, releasing numerous pro-hypertrophic, pro-fibrotic, and pro-inflammatory mediators. In addition, vasoactive hormones, various growth factors, cytokines, and the local renin-angiotensin system (RAS) act in an autocrine and/or paracrine mode. Collectively, the above-mentioned mechanisms orchestrate effects that contribute to pathological remodeling processes leading to cardiac hypertrophy, fibrosis, and inflammation. AT II: angiotensin II; CT-1: cardiotrophin-1; ECM: extracellular matrix; ET-1: endothelin-1; FGF-2: fibroblast growth factor-2; ICAM-1: intercellular adhesion molecule-1; IL-1: interleukin-1; IL-6: interleukin-6; NE: norepinephrine; TGF-ß: transforming growth factor-ß; TNFα: tumor necrosis factor-α.
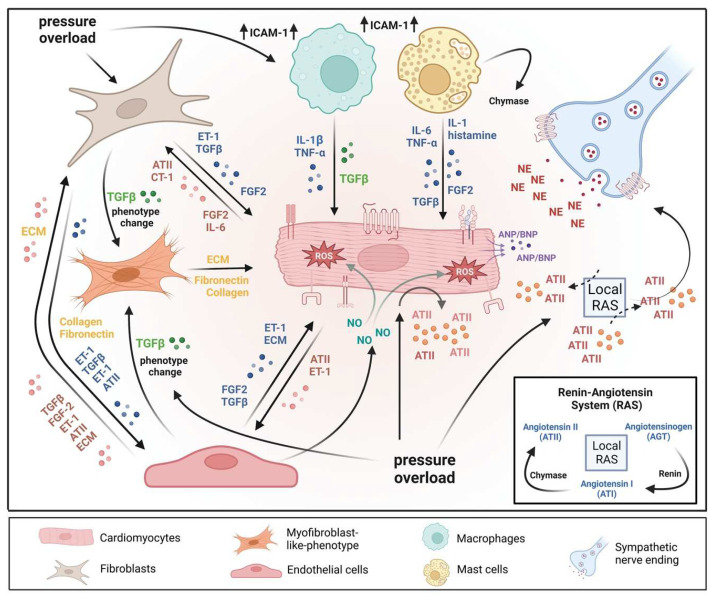


Major factors secreted by cardiac endothelial cells (Table 2) comprise nitric oxide (NO), endothelin 1 (ET-1), prostaglandin I2 (PI2), and angiotensin II (AT-II), which directly influence cardiac metabolism, growth, contractile performance, and rhythmicity of the adult heart [41]. In response to various stimuli, activated endothelial cells express adhesion molecules, including intercellular adhesion molecule-1 (ICAM-1) and vascular cell adhesion molecule-1 (VCAM), which attract and further promote the infiltration of immune cells into the myocardium (Figure 1). One major mediator produced and secreted by endothelial cells is NO (Table 2; Figure 1). Among the numerous functional influences of NO are cardiac-related functions, including key regulators of vasodilation, reduction of permeability and thrombogenesis, and inhibition of inflammation [42]. Another active mediator secreted by endothelial cells is CNP (Table 2). Together, NO and C-type natriuretic peptide (CNP) contribute to the suppression of cardiac hypertrophy by up-regulating cyclic GMP (cGMP)-dependent protein kinase 1 (PKGI) signaling [43] by inhibiting calcineurin (Figure 2).
Figure 2An overview of the pro-hypertrophic (**left panel**) and anti-hypertrophic (**right panel**) signaling pathways regulating the hypertrophic response in the cardiomyocyte. Increased intracellular Ca^2+^ levels mediated by TRPCs and Ca^2+^ import promote pro-hypertrophic transcriptional signaling events via calcineurin-NFAT and activation of PKC. PLC may contribute to these axes in the activation of alpha-adrenergic receptor signaling. Although canonical MAPK signaling via RTKs including FGFR-1 promotes pro-hypertrophic signaling, the PI3K-AKT axis plays an opposing role in hypertrophic signaling via inhibition of GSK3β and activation of YAP transcriptional activity. Increased secretion of cytokines promotes transcriptional activation of the pro-hypertrophic gene program in the nucleus not only via JAK-STAT but also the MEK5-ERK5 axis. On the other hand, increased pressure overload in cardiac tissue promotes secretion of ANP and BNP by cardiomyocytes, leading to vasodilation and an anti-hypertrophic response in cells via an increase in intracellular cGMP levels, which leads to activation of PKG, in turn mediating reduced hypertrophic growth. Activation of JNK and p38 stress signaling events in the cardiomyocytes, although leading to cardiomyopathy and heart failure, results in inhibition of NFAT through phosphorylation events that prevent its nuclear localization and pro-hypertrophic transcriptional regulation activities, thereby blocking the calcineurin axis. Increased secretion of TGF-β during increased pressure stress can lead to mixed responses, with canonical TGF-β-SMAD2/SMAD3 signaling leading to anti-hypertrophic responses, whereas activation of non-canonical SMAD1/SMAD5 leads to pro-hypertrophic responses.
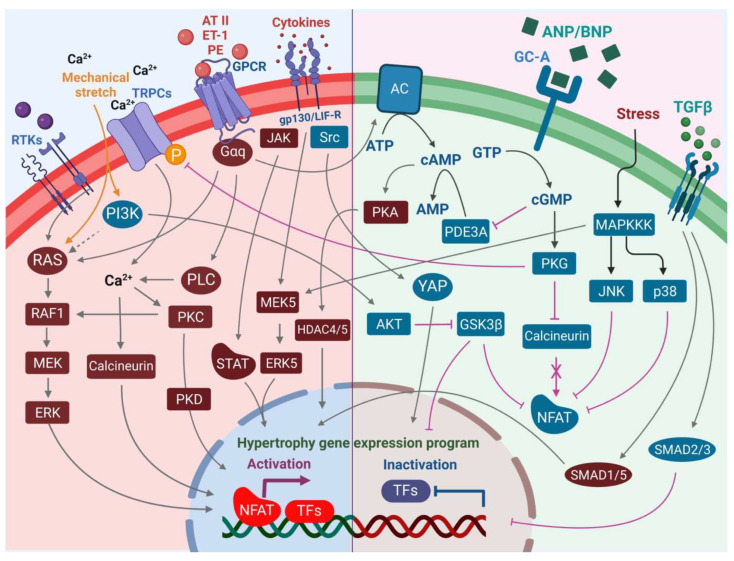



Another endothelium-derived factor next to NO and CNP is ET-1 (Table 2) [44], which contributes to cardiac hypertrophy and fibrosis as a major growth factor. Aside from endothelial cells, ET-1 is, amongst others, also expressed in non-endothelial cells, such as fibroblasts and cardiomyocytes (Table 2; Figure 1). Functioning in an autocrine and paracrine manner, ET-1 seems to have important effects during the development of cardiac hypertrophy [45]. ET-1 exhibits a positive inotropic effect as well as triggers cardiac hypertrophy responses [46]. Moreover, cardiac endothelial cells carry enzymes with protease activities, like the angiotensin-converting enzyme (ACE) and chymase (Table 2; Figure 1), which may contribute to changes in local levels of AT-II [47]. Besides fibroblasts, endothelial cells may also contribute to cardiac fibrosis (Figure 3). For example, it is known that endothelial cells and pericytes as the capillary lining cells wrapped around them, control cardiac fibroblast numbers [38]. Whether this contribution is similarly relevant as the proliferation and activation of resident fibroblasts upon exposure to pressure overload await further investigation [29].
Figure 3**Schematic illustration of the process of fibrotic scar formation at the cellular level.** The myocardium develops cardiomyocyte hypertrophy under pressure overload conditions, triggering concomitant inflammatory processes and fibrotic scar formation. The evidence discussed in the text suggests a central role for resident fibroblasts, nonetheless cardiac endothelial cells may also contribute to myofibroblast-like cells and drive cardiac fibrosis. Resident and infiltrating immune cells, including mast cells, macrophages, and neutrophils, enhance this phenotype change by releasing TGFß while mediating tissue inflammation via cytokines such as TNFα, IL-6, and IL-1. These mechanisms increase the number of myofibroblasts and the accumulation of collagen, which accelerates fibrotic scar formation in the microenvironment of cardiac hypertrophy.
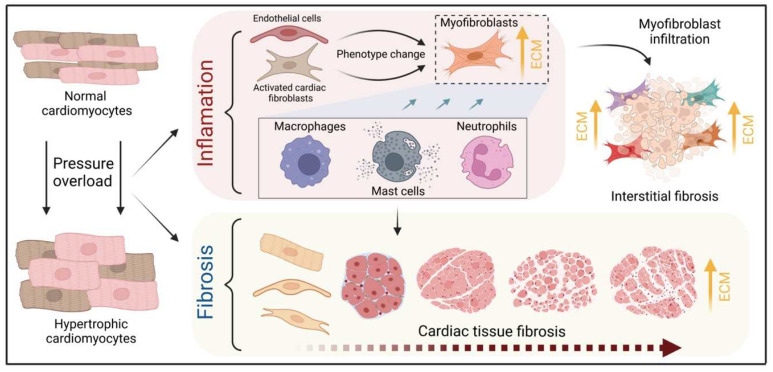


## 3. The Role of Immune Cells in Cardiac Hypertrophy

The pathogenesis of pressure overload and heart failure has been suggested to be in close context with the activation of inflammatory cells and the release of inflammatory mediators (Figure 1) [48]. Early inflammation in hypertrophic cardiomyopathy (HCM) may be brought on by cardiomyocyte disorganization, sarcomere damage, mitochondrial oxidative stress, and microvascular dysfunction [49]. Numerous investigations have shown that HCM patients have leucocyte infiltration in the myocardium and elevated levels of inflammatory cytokines, which may be crucial to the condition of HCM and its development to the dilated-phase end stage [50,51,52]. Additionally, genetic deletion of IL-6 has been shown to mitigate TAC-induced LV dysfunction and hypertrophy, demonstrating a crucial role for IL-6 in the pathophysiology of LV hypertrophy in response to pressure overload [53]. The nodal point for integrating hypertrophic and inflammatory signals in the myocardium is CaMKII, whose activity is elevated in MI hearts and which promotes cardiac hypertrophy and inflammation, processes that are persistently stimulated by cardiac injury [54].

### 3.1. Cardiac Mast Cells

Identification of the presence of mast cells in the heart tissue of animals and humans [55,56], as well as the discovery of mast cells as the source of an array of mediators (Table 2) [57], clearly emphasize the crucial participation of innate immune cells, especially cardiac mast cells, in cardiac hypertrophy and remodeling (reviewed in [48,58]).
cells-12-01780-t002_Table 2Table 2Mediators influencing the microenvironment in cardiac hypertrophy.Vasoactive PeptidesSecretion from/LocationReferencesAT-IIcardiomyocyte[59,60]AT-IIendothelial cell [41]ET-1cardiomyocyte[45,61] ET-1fibroblast[45,61,62,63]ET-1endothelial cell[41,45,61,64]**Catecholamines**

NEsympathetic nerve ending[65,66] **Growth factors**

FGF (aFGF, bFGF)cardiomyocyte[67,68] FGF (aFGF, bFGF)non-myocyte[69] FGF-2 (bFGF)fibroblast[67,68,70] FGF-2 (bFGF)endothelial cell [71,72,73]FGF-2 (bFGF)mast cell[57,74,75] High-FGF-2 (Hi-bFGF)fibroblast[70] TGF-ßcardiomyocyte[76,77,78]TGF-ßfibroblast[45,62,70,76,77,78]TGF-ßendothelial cell[76,77,78] TGF-ßmast cell[57,75,79,80] TGF-ßmyofibroblast[81,82,83] **Cytokines**

IL-6, CT-1, LIFcardiomyocyte [84,85,86,87,88] IL-6, CT-1, LIFfibroblast [85,86,87,88] IL-6mast cell [57,75] IL-1mast cell [57,75] TNFαmast cell [57,75,89,90,91,92] **Various other components**

VCAM-1, ICAM-1endothelial cell [42] ECM componentscardiomyocyte [93]ECM componentsfibroblast[93]ECM componentsendothelial cell [93]ECM componentsmyofibroblast[81,94] Histaminemast cell[55]Chemotactic factorsmast cell[57,75]**Anti-hypertrophic peptides**

ANP, BNPcardiomyocyte [8,43,95]NOendothelial cell [41,42]CNPendothelial cell [43]**Enzymatic activities**

Local RAScardiac tissue[96,97,98] AGT, renin, ACE, AT1, AT2cardiac tissue[99,100]AGTcardiomyocyte [101]AGTfibroblast[101]Reninmast cell [65,102]ACEendothelial cell [103]Chymase (alternative ACE)endothelial cell [103]Chymase (alternative ACE)mast cell [57,75,104] 


Activated cardiac mast cells were identified in spontaneously hypertensive rats as a major source of growth factors (Figure 1), such as TGF-ß and bFGF, in areas of myocardial fibrosis [105]. This is consistent with findings that the release of TGF-ß provokes an increase in collagen production alongside the differentiation of fibroblasts to myofibroblasts (Figure 1) [106], and indicates that cardiac mast cells also contribute to the key steps of cardiac tissue fibrosis [107]. Another major mediator that is released upon mast cell degranulation in the heart is histamine (Table 2) [55]. Histamine is a neurohormonal mediator that binds to histamine H1, H2, and H3 receptors, thereby inducing various cellular functions [108,109] as well as cardiac hypertrophy (Figure 1) [110]. Notably, cardiomyocytes express the histamine H2 receptor, which is coupled to the beta receptor and Gs proteins [111,112,113,114,115]. Consistently, histamine triggers positive inotropic effects [116,117]. In contrast, blocking the histamine H2 receptors decreases cardiac output [116]. The application of famotidine, a histamine H2 receptor antagonist, in chronic heart failure (CHF) patients, was found to decrease left ventricular remodeling [118].

Another characteristic of mast cells involves their strategic location often at a perivascular site, thereby exerting regulatory functions on endothelial cells. Mast cells synthesize several endothelial cell activators comprising, amongst others, the platelet-activating factor (PAF), IL-1ß, IL-4, and tumor necrosis factors alpha (TNFα) [119,120,121]. Several studies have indicated mast cell degranulation as a major source of TNFα (Table 2; Figure 1) [89,90,91,92]. Even though many cardiac cells have been described to generate TNFα, cardiac mast cells appear to constitutively express TNFα [89,91] and activate TNFα/nuclear factor kappa B (NF-κB)/IL-6 cascades [105]. Activation of the TNFα/NF-κB axis leads to the activation of p38-MAPK (Figure 2), collectively causing hypertrophy and dysfunction of the heart [122,123]. Moreover, mast cells release other cytokines including IL-1 and IL-6 (Table 2; Figure 1) [57,75]. IL-6 cytokine family binds the common co-receptor glycoprotein 130 (gp130) and thereby potentially takes an active role in cardiac hypertrophy induction via the JAK/STAT pathway (Figure 2) [124,125].

Although several studies suggest that cardiac mast cells are a source of renin, released upon mast cell degranulation (Table 2) [65,102], the major source of renin in the myocardial microenvironment is complex [126,127]. Mast cells release the proteolytic enzyme chymase, which catalyzes, independently of ACE, the conversion of AT-I to AT-II (Figure 1) [104]. Thus, mast cell renin and chymase may serve as an alternative way to upregulate AT-II levels in the myocardial microenvironment, and it has been demonstrated in rat models that mast cell inhibition using mast cell stabilizer cromolyn sodium reduces pathological left ventricular remodeling [128].

### 3.2. Monocytes & Macrophages

Healthy and injured cardiac tissues possess heterogeneous populations of macrophages, in both humans and mice (Figure 1) [129]. Most macrophages within the heart are established embryonically from the yolk sac and fetal liver progenitors, similar to tissue macrophages of the liver or brain. Local proliferation in contrast to monocyte recruitment serves to maintain resident macrophage subsets [130,131]. In the absence of disease, self-renewal serves to maintain local tissue macrophage populations [132]. Despite this, in response to pressure overload or ischemic injuries, the majority of macrophages are derived from the recruitment and differentiation of blood monocytes [133].

Cardiac macrophages are key effector cells mediating tissue remodeling and fibrosis (Figure 3) [134]. The initial and significant event for vascular lesion formation results from inflammatory cytokine- and growth factor-producing migrating macrophages (Figure 1) [135]. The accumulation of macrophages has been found in the perivascular space, where they co-localize with fibroblasts collectively producing collagen during cardiac hypertrophy (Figure 3) [136,137]. Consistent with this, other studies have found that pressure overload initiates endothelial cells of the intramyocardial arteries to exhibit intercellular adhesion molecule (ICAM)-1 and that accumulation of macrophages occurs adjacent to the ICAM-1 expressing arteries in the perivascular space (Figure 1) [138]. Additionally, vascular cells and monocytes synthesize and express monocyte chemoattractant protein (MCP)-1, a potent monocyte chemoattractant [139], primarily regulating the recruitment of macrophages to the vessels [140]. For example, the continuous infusion of the AT-II or norepinephrine in hypertensive rats demonstrated that MCP-1 induction was associated with adventitial macrophage accumulation in the aortic wall [139].

Collectively, this suggests that resident and recruited macrophages actively take part in the early responses to stress preceding hypertrophic remodeling.

### 3.3. Neutrophils

Under normal reparative conditions, neutrophil granulocytes are recruited to areas of acute inflammation, where they perform functions such as the clearance of dead cells and matrix debris (Figure 3) [141,142]. As key components of the inflammatory response, neutrophils also act on the recruitment, activation, and programming of antigen-presenting cells (APCs). Specifically, they attract monocytes and dendritic cells (DCs) by generating chemotactic signals, thereby influencing the differentiation of macrophages into a predominantly pro- or anti-inflammatory state [143,144,145]. Because neutrophil granulocytes are one of the most important cellular components of the body for the destruction of microorganisms, there is also the possibility that these cells damage host cells and tissues [146]. Accordingly, they may have deleterious effects on cardiac tissue when recruited to sites where pressure overload is present (Figure 3).

Several studies have reported that in response to hypertrophy triggered by pressure overload, the first leukocytes to appear in the myocardium within 3 days of injury are neutrophils (Figure 3) [147,148]. Activation of endothelial cells and subsequent expression of adhesion molecules allow the transmigration of neutrophils (Figure 3) [149,150]. In addition, inflammatory mediators such as TNFα, IL-1ß, and mast cell-derived histamine enhance this process [151,152,153]. Additionally, macrophage and neutrophil infiltration appeared in the first 3 days after injury next to ICAM-1 containing coronary arteries in the left and right ventricle, using a mouse model with inter-renal aortic banding. Moreover, these alterations of macrophage and neutrophil content occurred ahead in perivascular fibrosis (10 days), and cardiomyocyte hypertrophy (28 day) [154].

Neutrophils have been described to produce cytokines such as TNFα that drive macrophage and dendritic cell differentiation [143,145,155]. Additionally, neutrophilic nicotinamide adenine dinucleotide phosphate (NADPH) oxidase gets activated in response to pressure overload injury [156], resulting in the degranulation of neutrophils and thereby release of pro-fibrotic proteases (Figure 3) as well as reactive oxygen species (ROS) [157].

### 3.4. Lymphocytes

A growing body of research indicates that systemic inflammation may play a significant pathophysiologic role in the etiology of cardiac disease development, including HCM, and may have an impact on the severity of the phenotypic and clinical outcomes, including heart failure. A high neutrophil-to-lymphocyte ratio (NLR), a marker of oxidative stress damage, has been linked to an increased 5-year risk of sudden cardiac death associated with HCM [158,159], which supported further the prognostic significance of inflammation. In addition, the lymphocyte-to-monocyte ratio (LMR) and survival in patients with hypertrophic cardiomyopathy have been correlated, with a lower LMR being associated with a lower mortality rate [160].

However, in angiotensin II-induced HF models, the absence of B cells led to less hypertrophy and collagen deposition, the preservation of left ventricular function, and, in conjunction with these changes, a decrease in the expression of proinflammatory cytokines and apoptosis in the myocardium [161]. Different studies have also reported that activation of NK T cells improved cardiac remodeling events and failure in mice by increasing the expression of cardioprotective cytokines, including IL-10 [162,163]. Additionally, the activation of invariant natural killer T (iNKT) cells may act as a preventative measure against HF brought on by pressure overload as their disruption was shown to worsen cardiac hypertrophy [164]. Importantly, Ayach et al. emphasized the crucial role of c-KIT signaling in preventing ventricular dilation and hypertrophy as well as the maintenance of cardiac function after infarction by mediating the mobilization of NK and angiogenic cells derived from bone marrow, which helps with improved remodeling after MI [165].

On the other hand, a case study of a primary cardiac lymphoma (PCL) patient with T cell-lymphoma was shown to be in association with hypertrophic cardiomyopathy [166], besides a different patient with acute lymphoblastic leukemia (ALL) being reported in another case study to have developed HCM after cell therapy interventions using T cells [167]. Apart from these reports, in conclusion, the majority of the evidence points to lymphocytes having a favorable influence on HCM remodeling events.

### 3.5. Sympathetic Neurons

Sympathetic neurons that innervate the heart and release norepinephrine (NE) also express the endothelin receptor A (ET-A) [168,169]. ET-resulted in a tremendous NE release in cocultured cardiomyocytes and sympathetic neurons with exaggerated hypertrophy of cardiomyocytes compared to monocultured cardiomyocytes. In contrast, mice lacking the ET-A receptor exclusively in sympathetic neurons showed less adverse structural remodeling, and cardiac dysfunction when exposed to pathological pressure overload [170].

Substantial amounts of renin released in the cardiac microenvironment upon cardiac mast cell degranulation [65] result in both AT-II formation within striking distance of AT1 receptor-expressing cardiac sympathetic nerve terminals and enhanced NE release (Figure 1) and arrhythmias (Figure 4) [171,172]. The fact that these events can be prevented by mast cell-stabilizing agents confirms the central role of cardiac mast cell-derived renin in AT1 receptor signaling [65]. Locally produced AT-II thus activates the AT1 receptor at sympathetic nerve endings, resulting in increased NE release (Figure 1) [65].

## 4. Mediators of Cardiac Remodeling

Mechanical stretch and neurohumoral mechanisms identify the most proximal stimuli for initiating hypertrophic signaling pathways (Figure 1) [3]. Due to hemodynamic overload, cardiomyocytes undergo mechanical stress and thereby release autocrine and paracrine signaling factors, such as growth factors, hormones, cytokines, and chemokines (Table 2) [14]. Furthermore, mechanical stress is sensed by both cardiac fibroblasts resulting in the production and release of signaling mediators (Figure 1) [173], and cardiac endothelial cells, which communicate with cardiomyocytes by secretion of autocrine and paracrine mediators [43,174]. Cardiomyocytes sense these ligands through a multitude of G-protein-coupled receptors (GPCRs), growth factor receptors, and cytokine receptors (Figure 1 and Figure 2) [3]. Orchestrated mechanisms of the induction, maintenance, and progression of cardiac hypertrophy, particularly left ventricular hypertrophy, underlie a series of events that follows the activation of cardiomyocytes upon pressure overload/mechanical stress.

### 4.1. Activation of the Local Renin-Angiotensin System (RAS)

In addition to the classical circulating renin-angiotensin system (RAS) [96], the heart has a local RAS that mediates autocrine, paracrine, and intracrine effects (Figure 1) [97,98]. Components of the RAS, including angiotensinogen (AGT), renin, ACE, AT-I, and AT-II, are expressed in the heart [99,100], and component expression is upregulated in cardiomyocytes in vitro in response to stretch [175,176]. Several studies have indicated that hemodynamic overload activates the local RAS and highlighted the crucial role of the AT1 receptor in strain-induced cardiac hypertrophy [177,178,179,180]. Thus, mechanical stress can be considered the major upstream trigger that activates the local RAS and leads to increased AT-II levels throughout the microenvironment.

### 4.2. Reactive Oxygen Species (ROS)

Reactive oxygen species (ROS) such as superoxide anion (O^−^_2_), hydroxyl (OH), and hydrogen peroxide (H_2_O_2_), and reactive nitrogen species including nitric oxide (NO) and peroxynitrite (ONOO^−^) classify reactive species involved in redox signaling. The latter results from the reaction of (O^−^_2_) with NO [181]. Data suggest that both direct and indirect mechanisms resulting from redox signaling within and between endothelial cells and cardiomyocytes are responsible for functional communication between these cells [23]. Moreover, redox signaling not only influences many physiological processes in the heart but also plays an important role in pathological cardiac remodeling [182,183].

In cardiac cells, several sources of ROS have been described, such as mitochondria [184], xanthine oxidase (XO) [185], uncoupled NO synthases (NOS) [186], and NADPH oxidases (NOXs) [187]. The interactions of NOX proteins with NOS-derived NO have been highlighted to be particularly important for redox signaling in the development of heart failure (Figure 1) [187,188,189].

An increase in the cardiac generation of ROS and therefore an increase in oxidative stress has been implicated in pressure-overload-induced left ventricular cardiac hypertrophy (LVH) and heart failure (Figure 1) [13,190]. Additionally, the development of cellular hypertrophy and remodeling has been found to implicate increased ROS production, and activation of the mitogen-activated protein kinase (MAPK) superfamily, where redox-sensitive protein kinases, are known to be partly responsible. Moreover, cardiomyocyte apoptosis and necrosis may be due to increased oxidative stress (Figure 4), which is described to be associated with the transition from compensated pressure-overload-induced hypertrophy to heart failure. Furthermore, alterations in the redox-sensitive activity of several key proteins including sarcolemma ion channels and exchangers and sarcoplasmic reticulum calcium release channels, which collectively account for excitation-contraction coupling, contribute to myocardial contractile dysfunction (Figure 4). Beyond that, the consequent generation of peroxynitrite (ONOO^−^) as a result of increased inactivation of NO has been attributed to indirect effects of ROS, leading to coronary vascular endothelial dysfunction and peroxynitrite-induced inhibition of myocardial respiration [191].

Several mediators including AT-II, ET-1, alpha-adrenergic agonists, TNFα, and mechanical forces trigger NOX2 activation. Via induction of four cytosolic regulatory subunits (p47^phox^, p67^phox^, p40^phox^, and RAC1), these mediators initiate O_2_^−^ production [192], indicating that pressure overload subsequently increases O_2_^−^ levels (Figure 1). Excessive O_2_^−^ levels interact extremely rapidly with NO, resulting in peroxynitrite formation, thereby disrupting physiological NO signaling [189,193]. Hence, pressure overload shifts the balance towards increased ROS (Figure 1), a condition that suppresses the physiological functions of NO. Consistently, O^−^_2_ has long been recognized to be implicated in severe cardiovascular diseases. Moreover, reports indicate that NOS may generate O_2_^−^ instead of NO, a condition referred to as uncoupled NOS. The switch to O_2_^−^ generation appears as a consequence of tetrahydrobiopterin (BH4) depletion (usually through oxidation to BH2) or as NOS enzymes undergo post-translational modification [194]. Consistent, increased levels of O_2_^−^ and ONOO^−^ may be implicated in an amplifying mechanism that aggravates NOS uncoupling through oxidation of BH4 [195].

Hence as outlined above, reactive oxygen species should be considered as a group of key mediators driving pathological remodeling in the microenvironment of cardiac hypertrophy (Figure 1), especially regarding pressure overload.

### 4.3. Endogenous Storage Pools of AT-II in Secretory Granules

AT-II secretion into the culture medium upon mechanical stress of isolated cardiomyocytes has been observed and provides some evidence supporting the concept of increased local concentrations of AT-II [175]. Potential autocrine and paracrine regulatory mechanisms of AT-II may activate the AT1 receptor on cardiomyocytes and surrounding cells (Table 2) [196,197]. This in turn has been proposed to induce the release of autocrine and paracrine mediators, including vasoactive peptides, growth factors, cytokines, and ECM components, such as collagen (Figure 1) [45,62,70,198]. Potentiated or sustained AT1 receptor activation is likely associated with cardiomyocyte hypertrophy, fibroblast hyperplasia, and fibrosis (Figure 4) [59,199,200]. Alternative mechanisms have been proposed to contribute to the activation of the AT1 receptor upon binding of AT-II [201], including membrane stretch and mechanoactivation that can in turn promote distinct conformational rearrangements in the receptor, leading to alternative signaling outcomes [202,203]. Several proteins have been implicated as sensors of mechanical stretches, such as muscle LIM proteins, integrins, and their associated signaling pathways [204,205]. Network models have been developed to predict how these mechano-sensitive proteins work together to coordinate cardiomyocyte hypertrophy [206,207]. Mechanisms that integrate these events and propagate the stress signal to the AT1 receptor after activation by mechanical stress remain areas of active investigation. Interestingly, despite the absence of AT-II/AT1 signaling, cardiac hypertrophy, systolic dysfunction, and fibrosis occurred in response to pressure overload (Figure 4) [200].

### 4.4. The Two Faces of the TGF-ß Signaling

AT-II-activated fibroblasts release TGF-ß and ET-1 in a paracrine manner into cardiomyocytes, leading to hypertrophy (Table 2) [45]. Similar to mechanical stress, autocrine TGF-ß signaling promotes fibroblast proliferation and ECM production (Figure 1), especially collagen and fibronectin, whereas degradation of these components is reduced [208]. Several studies report that the canonical TGF-ß/SMAD2/3 signaling pathways (Figure 2) induce the expression of genes related to collagen, fibronectin, and other ECM proteins [209,210,211,212], which concomitantly contribute to cardiac fibrosis (Figure 1) [76]. Experiments using pressure-overload rats demonstrated that a TGF-ß neutralizing antibody inhibited fibroblast activation and proliferation, and diastolic dysfunction [76]. These data suggest TGF-ß as a central target and the inhibition of TGF-ß signaling as beneficial. In line with this, cardiac fibrosis was attenuated in SMAD3 deficient mice subjected to cardiac pressure overload, but interestingly cardiac hypertrophy and cardiac dysfunction were aggravated [213]. Also, another rat model revealed that worsened cardiac remodeling and increased mortality correlate with a reduction of ECM using a TGF-ß neutralizing antibody after myocardial infarction [214]. TGF-ß-activated kinase 1 (TAK1) binds directly to type II (TBRII) TGF-ß receptors. Identification of this interaction links TAK1 to the TGF-ß signaling cascade, implicating an additional way of hypertrophy induction in cardiomyocytes by TGF-ß signaling [215]. Thus, aside from contributing to cardiac fibrosis, the non-canonical TGF-ß/TAK 1 signaling pathway has also been reported to promote cardiac hypertrophy (Figure 2) [216]. Altogether, TGF-ß is released from cardiomyocytes, fibroblasts, and endothelial cells in the healthy heart (Table 2) [77,78] and in the context of injury and repair also from myofibroblasts and infiltrating immune cells [82,83]. Thus, TGF-ß seems to be involved in adaptive or maladaptive processes most likely depending on the context, and may locally trigger interactions between different cell types such as cardiomyocytes and fibroblasts (Figure 1) and thereby impact cardiac hypertrophy, fibrosis, and the development of heart failure (Figure 4).

### 4.5. Endothelin-1 Effects

Endothelin-1 (ET-1) is an endothelium-derived vasoconstrictor of 21 amino acids. Later, two additional homologs (ET-2 and ET-3) were identified. ET-1 is released from vascular endothelium and other cells including cardiomyocytes (Figure 1) after cleavage from a large precursor peptide [217]. ET-1 is the predominant endothelin in the heart and is identified as a potent hypertrophic stimulus in neonatal cardiomyocytes [218]. ET-1 is a ligand for two GPCRs: ET-A and ET-B where 90% of the endothelin receptors on cardiomyocytes belong to the ET-A subtype (Figure 2) [219]. In rat hearts, the ET-A is predominant and identified to be coupled to both the Gq and Gi subfamily of G-proteins (Figure 2) [220,221]. In addition, a characteristic pattern of gene expression is induced by ET-1 in ventricular neonatal rat cardiomyocytes (NRC) including immediate early genes (c-FOS, c-JUN, EGR-1), early genes (ANF, β-MHC, α-sk actin), and later on, ventricular MLC-2 and α-cd actin [222]. The Gq-RAS-RAF-ERK pathway may be involved in these transcriptional changes (Figure 2). Furthermore, ET-1 activates the Ras-MEKK1-SEK-JNK pathway contributing to the hypertrophy-associated gene expression program field [223].

ET-1 causes cell damage in cardiomyocytes in vivo, and experiments with long-term treatment with the ET-A receptor blocker BQ-123 showed improved survival of rats with heart failure [224].

The release of ANP and BNP from cardiomyocytes can also be triggered by AT-II and ET-1, though cardiomyocyte stretch is the main regulatory mechanism for ANP and BNP production [225].

### 4.6. FGF-2 Effects in Scar Formation

In general, considering the epigenetic state and very low proliferative potential of adult cardiomyocytes, consensus exists that there is only a small ability to regenerate injured myocardium through the proliferation of cardiomyocytes [226,227]. Instead, scar formation occurs through infiltrating highly proliferative cardiac fibroblasts (Figure 1 and Figure 3) [228]. A key player is FGF-2 (bFGF), which is expressed by numerous cell types in the adult myocardium. FGF-2 is released upon cardiac injury from its “storage site” (Table 2) thereby potentially activating cell surface receptors, such as FGFR (Figure 2) [229]. Moreover, AT-II, ET-1, and FGF-2 itself are known to promote FGF-2 gene expression (Table 2) [67,230]. Accordingly, FGF-2 increases both fibroblast and myofibroblast proliferation [231], therefore contributing to both enhanced scar formation and stiffness during cardiac injury (Figure 3). Noteworthy, FGF-2 exists as an isoform with a high molecular weight (Hi-FGF-2) and low molecular weight (Lo-FGF-2), thus it is important to determine the potential effects of both in the context of cardiac hypertrophy and tissue remodeling. In the past, several in vitro studies revealed evidence for an important role of FGF-2 in cardiac hypertrophy (Figure 1). Consistent with reports Lo-FGF-2 alters the gene profile of contractile proteins from “adult” to “fetal” programs when added to cultured neonatal cardiomyocytes, a distinct characteristic that is attributed to pressure overload-induced cardiac hypertrophy in vivo. Although data seems contradictory as others reported that cardiomyocyte hypertrophy is stimulated only by Hi-FGF-2, both in vivo and in vitro [198,232]. Hi-FGF-2 accumulates preferentially in response to stress stimuli (Figure 1), including AT-II [233] and oxidative stress [234]. This is further supported by others who found that Hi-FGF-2 is preferentially accumulated and released by cardiac fibroblasts which induce paracrine cardiomyocyte hypertrophy (Table 2) [70]. Once released, Hi-FGF-2 may directly interact and activate the tyrosine kinase receptor FGFR-1 (Figure 2) [235], and downstream MAPK signaling [70,236]. Lo-FGF-2 exhibits cardioprotective effects, especially against post-ischemic cardiac dysfunction [237]. One mechanism for the effects of Lo-FGF-2 is its potent angiogenic activity that may increase resistance to ischemic injury and cardioprotection [67,238,239]. In conclusion, these data imply that Hi-FGF-2 is a contributor to cardiac hypertrophy, fibrosis, and heart failure (Figure 4), while Lo-FGF-2 seems to exert opposite functions as a component of adaptive responses in the injured myocardium [240].

FGF-2 null mice had a marked reduction of the hypertrophic response in cardiomyocytes in response to pressure overload [241]; however, questions remain whether the entire blockade of FGF-2 signaling is therapeutically beneficial. Considering data highlighting the role of FGF-2 in the progression of many cancer types [242,243,244,245,246], blocking of FGF-2 may have beneficial effects as shown in reports on the elimination of tumor angiogenesis [247]. But, in the context of ischemic heart disease, inhibition of FGF-2 signaling may be detrimental, since an angiogenic effect by Lo-FGF-2 upregulation may be desirable [67,238,239]. Although data suggests functions for Hi-FGF-2 and Lo-FGF-2 in the myocardium, further investigations are certainly needed to understand (a) the precise outcomes of targeting one or the other isoform, (b) the effects on exact organs/cells, and (c) to define the precise function of the isoforms in the context of cardiomyocyte hypertrophy and fibrosis. Additionally, unwanted effects of Hi-FGF-2 and Lo-FGF-2 need to be considered. Moreover, in addition to FGF-2, TGF-ß, AT-II, catecholamines, and other molecules as well orchestrate the response to hemodynamic stress (Figure 1), which suggests that targeting just one mediator may not be sufficient.

ERK as the most prominent downstream effector of FGF-2 signaling plays a predominant role in the development of both physiological and pathological cardiac hypertrophy (Figure 2). While cytosolic functions of ERK upon activation through pressure overload and mediators are believed to promote the development of physiological hypertrophic conditions, nuclear transcriptional activations mediated by ERK promote a pathological hypertrophic response in CMs (Figure 2) [248,249]. Hypertrophic stimuli such as AT-II, ET-1, cytokines, catecholamines, and biomechanical stress may also contribute to detrimental ROS formation in cardiomyocytes, and additional autophosphorylation of ERK1/2 has been reported to trigger pathological ERK1/2-mediated cardiac hypertrophy (Figure 2) [250,251]. These changes can then activate several hypertrophic signaling mediators regulated by ERK1/2 [249,252].

Hyperactivation of ERK1/2 activity is most frequently linked to HCMs caused by genetic abnormalities [253,254]. While genetic variant-induced hyperactivation of ERK is closely linked to pathogenic remodeling, normalization of ERK activation by simvastatin treatment restores contractility and protects against fibrosis in animal models [255,256].

One study reported different cardiac hypertrophic responses using both mice that completely lacked ERK1/2 protein in the heart and mice that expressed an activated MEK1 in the heart. Inhibiting MEK-ERK1/2 in mice lacking ERK1/2 in the heart causes eccentric cardiac growth with elongated cardiomyocytes, whereas activation of MEK1-ERK1/2 signaling by the overexpression of an active MEK1 mutant appears to be responsible for the concentric type of hypertrophy with thicker cells [257]. Thus, increased pre- versus afterload have been described to result in typical hypertrophic responses, and ERK1/2 seem to exhibit a central role, partially regulating the underlying molecular mechanisms. Induction of ERK1/2 translocation to the nucleus in adult rat myocytes, corresponded to reduced myocyte lengths and increased width, under both baseline and chronic pacing conditions [258], pointing to the critical role played by ERK signaling in balancing concentric and eccentric hypertrophic growth (Figure 2).

### 4.7. Cytokines and Inflammasome in Cardiac Remodeling

Cytokines of the interleukin-6 (IL-6) family are key molecules for the local regulation of hypertrophic responses in cardiomyocytes (Figure 1). Pressure overload acts as a strong trigger for the upregulation of genes related to leukemia inhibitory factor (LIF) and cardiotrophin-1 (CT-1) in the adult human myocardium [259,260]. Cardiomyocytes and cardiac fibroblasts produce leukemia LIF and CT-1 (Table 2) [261]. The release of Hi-FGF-2 from cardiac fibroblasts (Table 2) has been suggested to act in an autocrine way and trigger the release of pro-hypertrophic CT-1 [70,262]. Moreover, cardiomyocytes also express autocrine-acting CT-1, and CT-1 induces hypertrophy of cardiomyocytes in vitro [263]. Increased production and release of LIF, CT-1, and IL-6 in cardiac fibroblasts in response to AT-II can contribute to cardiomyocyte hypertrophy by paracrine activation of the gp130-linked downstream signaling (Figure 2) [264]. Interestingly, IL-6 contributes to the induction of massive collagen release by cardiac fibroblasts in response to AT-II and norepinephrine stimulation [265,266], consistent with a pro-hypertrophic response. Alternatively, LIF stimulates several beneficial effects including reduction of collagen production and matrix metalloproteinase activity in cardiac fibroblasts, resulting in an inhibition of differentiation of cardiac fibroblast to myofibroblast [267]. Likewise, the role of CT-1 seems unclear as consistent with reports describing CT-1 as having a potent hypertrophic effect on cultured cardiomyocytes [268] in addition to cardioprotective effects such as promoting cardiomyocyte survival [269]. In conclusion, during the process of developing cardiac hypertrophy, cytokine release is increased in response to a variety of stress stimuli, including pressure overload, injury, and mediators like AT-II. However, since IL-6 has a negative inotropic effect, its function is still unclear, suggesting the possibility of detrimental impacts by IL-6 driving hypertrophy toward heart failure.

According to data binding of all IL-6-type cytokines to their common receptor subunit gp130 potently activates STAT3 and to a lesser extent STAT1 (Figure 2) [270]. Transgenic mice with cardiac-specific STAT3 over-expression found that STAT3 holds a key role in hypertrophic and protective signaling, respectively. STAT3 induced the expression of cardiac protective factors and guarded against decreases in the expression rates of cardiac contractile genes in the case of doxorubicin-induced cardiomyopathy [271]. In line with this, another study that used pressure overload on ventricular-restricted gp130 receptor knockout mice found a rapid onset of dilated cardiomyopathy and induction of cardiomyocyte apoptosis. In comparison, a normal cardiac structure and function were found under basal conditions, and compensated hypertrophy was found in control mice under pressure overload [272]. These observations suggest a key role of the gp130/STAT pathway in cardiomyocytes for transmitting adaptive and protective functions in response to pressure overload and injury. However, a study on transgenic mice that expressed a dominant negative mutant of gp130 (to decrease activation of this pathway) reported concomitant to a suppressed STAT3 activation a significantly smaller hypertrophic response when subjected to pressure overload [273], suggesting a pro-hypertrophic function for STAT3. Whether the effects of the gp130 signaling pathway are beneficial or detrimental remains unclear. Since pressure overload triggers hypertrophic responses in cardiomyocytes via GPCRs in turn activating PKC and PKD (Figure 2) [274], potential crosstalk of signaling pathways could be involved. Likewise, neonatal rat cardiomyocytes showed that stretch induces a transient activation in a sequential time order on PKC and other downstream targets as the successive components of the MAPK signaling cascade (Figure 2) [275].

In contrast, YAP1, a downstream effector of Hippo signaling regulating proliferation, survival, and organogenesis in mammalian cells, that can also be activated through SRC-mediated gp130 activation in cardiomyocytes [276], is involved in cardio-protective mechanisms against pressure overload stimulation of cardiac hypertrophy (Figure 2). Under chronic pressure overload conditions, activation of YAP transcriptional activity reduces the development of cardiac hypertrophy. Additionally, apoptosis and fibrosis effects on cardiomyocytes that can be prerequisites for myocardial infarction are reduced [277]. The transcriptional activity of YAP mediates compensatory cardiac hypertrophy under pressure overload conditions [278] to stop the progression of wall stress into myocardial infarction, while CMs are driven toward heart failure by the detrimental effects of YAP signaling loss-of-function [279].

Concomitant hypertrophic responses via activation of PKC and MAP kinases can also be triggered by AT-II (Figure 2). Cardiomyocytes under mechanical stress secrete AT-II [280]. Here, active PKC, with its numerous nuclear and cytosolic substrates, specifies the extensive crosstalk of signaling pathways in response to pressure overload. The alpha-isoform of PKC directly activates RAF1 [281], providing evidence for a complex link between the signaling pathway downstream of growth factor receptors in the context of cardiac hypertrophy. Others have reported that GPCR signaling can be linked directly to RAS GTPase (Figure 2) [282], and GTP-bound RAS interacts with many downstream effectors which in turn transmit the signal for activating multiple signaling pathways [283,284], potentially promoting hypertrophic responses in cardiomyocytes. Additionally, others reported that the C-terminus of the AT1 receptor associates with JAK2 upon binding of the ligand, resulting in JAK2/STAT3 pathway activation [285,286], indicating another example of the crosstalk of signaling pathways in response to hypertrophy-associated stress signals. These lines of evidence indicate that the discrepancy of data regarding the gp130 signaling pathway may be due to the extensive crosstalk between intracellular signaling pathways (Figure 2). Taken together, there are contradictory reports regarding the individual effects of IL-6, LIF, CT-1 and their signaling via the gp130 pathway in cardiac hypertrophy, thus further investigation is necessary for elucidating the exact mechanisms.

### 4.8. Calcineurin/NFAT in Cardiac Hypertrophy

Calcineurin as a Ca^2+^-dependent serine/threonine protein-phosphatase has been found to exhibit central pro-hypertrophic functions in the myocardium (Figure 2) [287,288]. Calcineurin contains two subunits: the 57–61-kDa catalytic subunit (CnA) and the 19-kDa regulatory subunit (CnB). Activation of this dimeric protein occurs through direct binding of the Ca^2+^-saturated adaptor protein calmodulin [289]. The mammalian heart only expresses CnAα, CnAβ, and CnB1, although there are three genes including CnAα, β γ encoding for CnA, and two genes (*CnB1 and B2*) encode for CnB. Calcineurin becomes activated in response to increased Ca^2+^ levels, which enables binding to transcription factors of the nuclear factor of activated T cells (NFAT) family (Figure 2) [289].

Pro-hypertrophic gene expression is activated upon binding, and through dephosphorylation of conserved serine residues at the N-terminus of NFAT by calcineurin, resulting in NFAT translocating into the nucleus (Figure 2). Here, NFAT regulates the expression of cardiac genes via association with GATA4 and myocyte enhancer factor 2 (MEF2), which are also transcription factors [290,291]. Noteworthy, several studies indicate that NFAT transcription factors act as primary calcineurin effectors in the heart, as they have been identified as necessary and sufficient mediators promoting cardiac hypertrophy [287,290,292]. Moreover, cardiomyocytes contain structural proteins located in the repetitive Z-disc that have been found to regulate calcineurin in addition to the activation via increased Ca^2+^ [293,294].

GPCR stimulation with hypertrophic agonists, including AT-II and PE on cultured neonatal rat cardiomyocytes indicated an increase in calcineurin enzymatic activity, which was induced by increased calcineurin Aß (CnAβ) mRNA and protein, compared to CnAα or CnAγ [295]. By that, human hypertrophied and failing hearts (Figure 4) also exhibit increased calcineurin activity [296], as well as in ventricular muscle with exposure to AT-II, ET-1, and Urotensin II in human failing heart [297]. Significantly, hypertrophied hearts in rodents subjected to aortic banding displayed upregulated calcineurin activity [298,299] and profound cardiac hypertrophy with rapid progression to dilated cardiomyopathy, extensive fibrosis, congestive heart failure, and sudden death (Figure 4) were observed in active calcineurin expressing transgenic mice [292].

Upregulated NFAT activity has been observed upon both physiological stimuli (exercise training, growth hormone-IGF1 infusion) and pathological stimuli (pressure overload, myocardial infarction) (Table 1) [15]. In contrast, the hypertrophic response to pressure overload and GPCR agonists was impaired in a model of transgenic mice exhibiting a targeted inactivation of calcineurin Aβ [300] and in transgenic mice expressing a dominant negative form of calcineurin A [298]. Furthermore, cardiac hypertrophy was prevented in a model using pharmacological inhibition of calcineurin A activity on transgenic mice with constitutively active calcineurin A [289,292].

These lines of evidence taken together indicate that calcineurin/NFAT plays a major role in the conversion of pathogenic stimuli into pathological cardiac remodeling, suggesting it is a key target in the setup of clinical prevention of cardiac hypertrophic (Figure 4). But data seems contradictory, as a study reported accentuated hypertrophy, impaired histopathology as well as risk for early death when applying calcineurin inhibitors [301]. Thus, further investigation is necessary to clarify if calcineurin/NFAT could be considered as a key target.

### 4.9. ANP/BNP in Cardiac Hypertrophy

Development of pathological cardiac hypertrophy is frequently linked to increased mRNA expression of atrial natriuretic peptide (ANP) and B-type natriuretic peptide (BNP), according to studies in both human and animal models [302,303], as well as an increase in the plasma levels of ANP and BNP with the severity of heart failure. Under critical conditions, more BNP than ANP is secreted, largely in the ventricles and atria, respectively. However, as heart failure worsens, ANP is also secreted in the ventricles; for this reason, the ventricles are crucial locations for both BNP and ANP [304]. Both ANP and BNP, as well as their more stable cleavage products, NT-proANP and NT-proBNP, respectively, are efficient biomarkers in the clinical diagnosis and management of heart failure (Figure 4) [305,306].

Besides the physiological effects of ANP and BNP such as vasodilation, regulation of sodium reabsorption and water balance as well as inhibition of the renin-angiotensin-aldosterone (RAA) system, collectively directed towards responding to cardiac pressure and volume dynamics and suppression of heart failure [307,308], ANP/BNP causes the cGMP-dependent PKG to be activated (Figure 2), which in turn prompts the Ca^2+^/calmodulin-dependent endothelial nitric oxide (NO) synthase to aid in the production of more NO, which relaxes the vascular smooth muscle cells and lowers systemic blood pressure [307,309,310]. ANP/BNP and NO can also counteract NE effects on the size expansion of cardiomyocytes, presumably through the cGMP-PKG-mediated cardioprotective axis resulting in the reduction of NE-stimulated Ca^2+^ influx [309,311].

Moreover, while ANP and BNP expression is being regulated by pro-hypertrophic transcriptional activation of NFAT, on the other hand, ANP and BNP can counteract as negative regulators of hypertrophy by PKG-mediated inhibition of calcineurin to curb nuclear translocation of NFAT (Figure 2) [312,313,314].

## 5. Mathematical Modeling of Cardiac Remodeling

Many mediators and pathways implicated in cardiac hypertrophy hinder the field’s ability to integrate individual findings into a common framework. Mathematical modeling has also been useful for the elucidation of intracellular and intercellular signaling that controls cardiac functionality.

### 5.1. Computational Models of Cardiac Hypertrophy

Several computational models have been developed to address this, providing systems-level insight into how cardiac hypertrophy is regulated. In the first model of hypertrophic signaling, Cooling et al. examined the factors that control the kinetics of IP3 [315]. They found that ET-1 induced a much more sustained IP3 signal than AT-II, which was best explained by differences in receptor kinetics. To obtain a more global view of hypertrophic signaling, Ryall et al. used a logic-based modeling framework [316] to simulate 193 reactions integrated across 14 pathways [317]. Comprehensive knockout simulations supported the conclusion that RAS GTPase is the hub of a bow-tie control structure, which integrates signals from many receptors and stimulates hypertrophy through partially redundant MAPK pathways. This was validated in new experiments comparing the effects of inhibition of RAS GTPase, MEK, p38, and JNK [317].

While neonatal cardiomyocytes have been extremely useful in the study of cardiac hypertrophy, it is well-known that they show limited maturity compared to adult cells [318]. But quantitatively, to what extent are in vitro data predictive of in vivo cardiac hypertrophy? To address this question comprehensively, Frank et al. used the model of neonatal cardiomyocyte hypertrophy [317] to attempt to predict the in vivo hypertrophy for 52 cardiac-specific transgenic mice [319]. Strikingly, they found that the model correctly predicted 78% of cardiac outputs, including four double-transgenic mouse models. Differences between model predictions and in vivo experiments may indicate differences between in vitro and in vivo mechanisms or specific transgenic mice whose hypertrophic phenotype depends on specific contexts (e.g., hormones, genetic background).

Indeed, examination of context-dependent regulation can elucidate new aspects of signaling networks. Khalilimeybodi et al. developed a computational method called CLASSED to systematically revise the previous model of Ryall et al. [316] using context-dependent experimental data from 550 experimental data from 230 literature articles. Examining areas of a model-experiment disagreement using CLASSED, they identified the reactions that should be removed or added from the network. They also found new crosstalks between Gβγ and CaMKII or calcineurin, which were validated in neonatal cardiomyocytes [316]. Most recently, models of cardiomyocyte signaling are being incorporated into models of multiscale integration of mechanics and signaling in pressure overload, hormones [320], or pregnancy [321]. Recently, researchers have attempted to mature iPSC-CMs by prolonged culture duration, metabolic substrates, and mechanical and electrical stimulation to model HCM and measure cellular morphology, contractility, electrophysiological property, calcium handling, and metabolism [322,323]. Therefore, network model reparameterization for iPSC-CM will be advantageous from the perspective of translational applications. Cardiac hypertrophy is associated with increased ventricular arrhythmia [324]. Interestingly, several nodes in the signaling network of hypertrophy (such as CaMKII, PKA, and calcineurin) modulate the ion channels [325,326]. Therefore, the involvement of these node states in multiscale electromechanical models may predict the association of hypertrophy and arrhythmia.

### 5.2. Computational Modeling of Fibrosis

As illustrated in Figure 2, the complexity of intracellular networks often prohibits the identification of the signaling mechanisms that control cellular responses to biochemical or mechanical stimuli upon hypertrophy. To address this challenge, Zeigler et al. developed a logic-based differential equation model of the cardiac fibroblast signaling network, which was successfully validated against 80% of 41 papers from the literature not used in model development [327]. This model predicted that stretch-mediated myofibroblast activation was mediated not by any single path from integrins to α-SMA expression, but by an autocrine TGF-β autocrine loop. They validated this new prediction in new experiments by using a TGF-β receptor inhibitor to block cardiac myofibroblast activation in mechanically-restrained collagen gels [327]. This model was later extended to predict the in vivo fibroblast dynamics after myocardial infarction, predicting how IL-1 can paradoxically enhance collagen production through the above autocrine TGF-β loop but suppress it through activation of NFkB and BAMBI [328]. To make patient-specific predictions, Rogers et al. connected the logic-based model to transcriptional responses from valvular interstitial cells treated with patient serum samples [329]. They found that endothelin-1, IL-6, and TGF-β were most important for explaining patient-specific fibroblast activation.

To predict therapeutic approaches, the fibroblast network model was integrated with DrugBank to predict FDA-approved drugs that could be repurposed against cardiac fibrosis [330]. Interestingly, the combination drug Entresto (valsartan/sacubitril) was predicted to be particularly effective due to combined suppression of ERK through valsartan and enhancement of PKG through sacubitril [330]. This prediction was validated by independent studies showing that Entresto decreases fibrosis due to pressure overload in rats [331,332] and heart failure in humans [333]. Watts et al. extended this model with estrogen signaling, predicting that the effects of some drugs may be sex specific [334], building on previous experimental studies of how estrogen affects cardiac fibroblast signaling and activation [335]. Others have combined network modeling with an experimental drug screen through machine learning to identify pathways by which drugs regulate new fibroblast phenotypes such as stress fiber organization [336].

While fibroblasts play a central role, they regulate fibrosis through communication with many other cell types. Jin et al. developed a mathematical model that simulated the communication between monocytes, macrophages, and fibroblasts that lead to fibrosis after myocardial infarction [337]. Their model was validated against dynamics of inflammation and collagen content after myocardial infarction and then applied to predict how the strength or timing of perturbations to TGF-β or MMP9 can modulate the kinetics of post-MI fibrosis. Using a similar paradigm, Chowkwale et al. developed a model of the post-MI communication between neutrophils, monocytes, macrophages, cardiomyocytes, and fibroblasts [338]. They validated the model against 61 of 84 experiments not used to build the model. Using this model, they identified key dynamic features that control inflammation, fibrosis, and a new concept of inflammation-fibrosis coupling. Specifically, they predicted that inflammation is amplified by positive feedback between neutrophils and IL-1β, macrophage phagocytosis of cardiomyocytes is critical for inflammation to drive fibrosis, and that fibroblast proliferation acts as an ultrasensitive switch to amplify collagen deposition [338]. Intriguingly, this dual-amplification control system identified for inflammation-fibrosis coupling [338] appears analogous to that in excitation-contraction coupling [339]. Intercellular models illustrate complex dynamic relationships that should be experimentally validated for potential therapeutic strategies.

## 6. Concluding Remarks and Future Directions

The microenvironment involved in the development of cardiac hypertrophy involves cardiomyocytes and non-myocardial cells, and the accompanying release of numerous pro-hypertrophic, pro-fibrotic, and pro-inflammatory mediators facilitating reciprocal interactions.

Cardiac fibroblasts are the main players in the development of fibrosis, nevertheless, endothelial cells that can undergo EndMT toward a myofibroblast-like phenotype are closely involved as well. Resident and infiltrating immune cells (mast cells, macrophages, neutrophils) enhance these processes while simultaneously contributing to tissue inflammation. Thus, considering all these mechanisms in the hypertrophic microenvironment, tailoring an efficient treatment regimen appears extremely complex. Sophisticated strategies and most likely multidirectional approaches are needed and should be well-approachable using computational modeling systems that allow the integration of all signaling components.

Since it is not feasible to discuss every cellular and molecular process involved in the development of different types of cardiac hypertrophy, we aimed to outline the main drivers of the hypertrophic microenvironment and the respective signaling pathways being affected. A necessary future approach will be the identification of the precise involvement of different cell types, cellular mediators released by them, and the respective activation of second messengers. This will allow us to evaluate the known and thus far unrecognized molecular signaling axes during disease development. Moreover, such data collection within a computational model will help to guide effective and selective targeting strategies in cardiac hypertrophy. Given the high prevalence of heart disease in the Western world, an important future effort should be to translate the knowledge gained into new pharmacological targets that help to delay or even stop the remodeling process and the severe consequences that patients experience after diagnosis of diastolic or systolic dysfunction.

## Figures and Tables

**Figure 4 cells-12-01780-f004:**
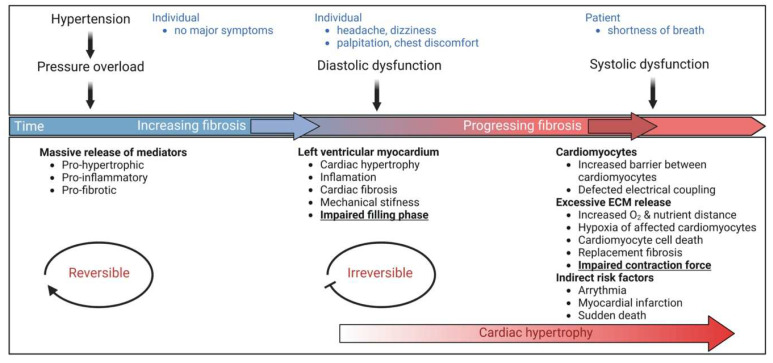
Schematic representation of the effects of changes in the microenvironment on cardiac function. Hypertension, a common cardiovascular disease, causes pressure overload followed by a massive release of pro-hypertrophic, pro-fibrotic, and pro-inflammatory mediators. At this stage, when individuals do not experience symptoms, hypertension, and its accompanying microenvironmental complications may be reversible with strategies such as lifestyle modification, however without any intervention, this could evolve into cardiac hypertrophy and fibrotic remodeling. Increasing fibrosis leads to mechanical stiffness and impaired filling phase, both prominent features of diastolic dysfunction. Common symptoms include headache, dizziness, palpitations, and chest discomfort. Notably, this phase is not reversible and requires pharmacological management. Late diagnosis or inadequate treatment leads to progressive fibrosis and detrimental changes at the molecular level, such as a barrier between cardiomyocytes at the cellular level, impaired electrical coupling, and hypoxia of affected cardiomyocytes, collectively resulting in cardiomyocytes’ cell death. The subsequent decreased contractile force characterizes systolic dysfunction while having severe consequences as individuals suffer from shortness of breath. Biomarker identification in a diagnostic screening approach could help detect early onset diastolic dysfunction in affected individuals, setting the platform for early management and preventive course of action to avoid the subsequent detrimental outcomes of the developing condition.

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
