# Peer review of "The Microenvironment of the Pathogenesis of Cardiac Hypertrophy"

_cells, 2023, doi:10.3390/cells12131780_

Round 1

Reviewer 1 Report

This is an excellent review article. I commend the authors for their effort and the compilation of research on cardiac hypertrophy worldwide in the last three decades. Though I have a few comments, none would significantly improve this review's quality except the following one: I recommend the authors include more recent research on cardiac hypertrophy (2015 to present). About 90% of the research work outlined in the review was done before 2015, which has already been compiled in several studies.

Author Response

This is an excellent review article. I commend the authors for their effort and the compilation of research on cardiac hypertrophy worldwide in the last three decades. Though I have a few comments, none would significantly improve this review's quality except the following one: I recommend the authors include more recent research on cardiac hypertrophy (2015 to present). About 90% of the research work outlined in the review was done before 2015, which has already been compiled in several studies.

Response: My coworkers and I thank you for reviewing our manuscript and for highlighting this point about the manuscript. In order to update the research references in the manuscript, collectively more than 100 changes have been made to the list of references in the revised manuscript, including removed and updated, or newly added studies with the majority of the updated references corresponding to the research published in the last 3 years.

Reviewer 2 Report

This is a thorough review regarding the regulation of pathological cardiac hypertrophy and fibrosis by the microenvironment.  The manuscript is well written and I have only a few minor suggestions:

- In the General Introduction, it might be beneficial to discuss the relative lack of cardiomyocyte proliferation in the remodeling process and potentially contrast this to skeletal muscle growth.

- Consider adding inflammation/immune differences between physiological and pathological hypertrophy in Table 1.

- The authors provide a good discussion of several inflammatory/ immune cell types.  While the literature is not as extensive, it would be beneficial to include a summary of lymphocyte involvement in hypertrophy.

- Suggest adding some of the work by the authors and others on computational modeling fibroblasts and fibrosis in the section on modeling hypertrophy.

Author Response

This is a thorough review regarding the regulation of pathological cardiac hypertrophy and fibrosis by the microenvironment.  The manuscript is well written.

Response: My coworkers and I thank you for reviewing our manuscript and for your valuable comments, the consideration of which greatly improved the overall quality and readability of the manuscript. We have revised the manuscript accordingly, point by point.

I have only a few minor suggestions

- In the General Introduction, it might be beneficial to discuss the relative lack of cardiomyocyte proliferation in the remodeling process and potentially contrast this to skeletal muscle growth.

Response: We have accordingly added the paragraph highlighted below to the introduction section:

While hypertrophy, or the enlargement of individual muscle fibers, is the primary mechanism by which skeletal muscle mass increases during postnatal development and a similar process may be induced in adult skeletal muscle in response to contractile activity, satellite cell proliferation and fusion, which increases the number of myonuclei, may also play a role in muscle growth during early but not late stages of postnatal development and in some forms of muscle hypertrophy 1.

Likewise, the number of endothelium and mesenchymal cells can also increase from birth through early adulthood, but on the other hand the entire complement of cardiomyocytes is created during pregnancy and remains constant throughout the human lifespan. Early infancy has the highest levels of cardiomyocyte exchange, which steadily declines during life to 1% per year in maturity in processes that increase very modestly in the vicinity of cardiac injury 2,3. Important to note that cell duplication is not always a result of cardiomyocyte cell cycle activity. In contrast, multinucleation and polyploidization occur during various phases of development including heart development as a result of premature cell cycle exit 4.

- Consider adding inflammation/immune differences between physiological and pathological hypertrophy in Table 1.

Response: We updated the Table and added to the text:

Early inflammation in HCM may be brought on by cardiomyocyte disorganization, sarcomere damage, mitochondrial oxidative stress, and microvascular dysfunction 5. Numerous investigations have shown that HCM patients have leucocyte infiltration in the myocardium and elevated levels of inflammatory cytokines, which may be crucial to the condition of HCM and its development to the dilated-phase end stage 6-8. Additionally, genetic deletion of IL-6 has been shown to mitigate TAC-induced LV dysfunction and hypertrophy, demonstrating a crucial role for IL-6 in the pathophysiology of LV hypertrophy in response to pressure overload 9. The nodal point for integrating hypertrophic and inflammatory signals in the myocardium is CaMKII, whose activity is elevated in MI hearts and which promotes cardiac hypertrophy and inflammation, processes that are persistently stimulated by cardiac injury 10.

- The authors provide a good discussion of several inflammatory/ immune cell types.  While the literature is not as extensive, it would be beneficial to include a summary of lymphocyte involvement in hypertrophy.

Response: We appreciate for bringing up this point. Lymphocyte involvement in the pathogenesis of hypertrophic cardiomyopathy is not solidly demonstrated, as the reviewer has noted. However, we would like to add the following section to provide a summary of some previously reported, mainly indirect, possible impacts of lymphocytes, including T cells, B cells, and natural killer (NK) cells in relation to pathogenic events in cardiac hypertrophy:

A growing body of research indicates that systemic inflammation may play a significant pathophysiologic role in the etiology of cardiac disease development, including HCM, and may have an impact on the severity of the phenotypic and clinical outcomes, including heart failure. A high neutrophil-to-lymphocyte ratio (NLR), a marker of oxidative stress damage, has been linked to an increased 5-year risk of sudden cardiac death associated with HCM 11,12, which supported further the prognostic significance of inflammation . In addition, the lymphocyte-to-monocyte ratio (LMR) and survival in patients with hypertrophic cardiomyopathy have been correlated, with a lower LMR being associated with a lower mortality rate 13.

However, in angiotensin II-induced HF models, the absence of B cells led to less hypertrophy and collagen deposition, the preservation of left ventricular function, and, in conjunction with these changes, a decrease in the expression of proinflammatory cytokines and apoptosis in the myocardium 14. Different studies have also reported that activation of NK T cells improved cardiac remodeling events and failure in mice by increasing the expression of cardioprotective cytokines, including IL-10 15,16. Additionally, the activation of invariant natural killer T (iNKT) cells may act as a preventative measure against HF brought on by pressure overload as their disruption was shown to worsen cardiac hypertrophy 17. Importantly, Ayach et al. emphasized the crucial role of c-KIT signaling in preventing ventricular dilation and hypertrophy as well as the maintenance of cardiac function after infarction by mediating the mobilization of NK and angiogenic cells derived from bone marrow, which helps with improved remodeling after MI 18.

On the other hand, a case study of a primary cardiac lymphoma (PCL) patient with T cell-lymphoma was shown to be in association with hypertrophic cardiomyopathy 19, besides a different patient with acute lymphoblastic leukemia (ALL) being reported in another case study to have developed HCM after cell therapy interventions using T cells 20. Apart from these reports, in conclusion, the majority of the evidence points to lymphocytes having a favorable influence on HCM remodeling events.

- Suggest adding some of the work by the authors and others on computational modeling fibroblasts and fibrosis in the section on modeling hypertrophy.

Response: We thank the reviewer for this nice suggestion. We have added a new section to the manuscript highlighting the latest works in literature focused on this point under a new heading “Computational modeling of fibrosis”:

5.2. Computational modeling of fibrosis

As exhibited in Figure 2, the complexity of intracellular networks often prohibits identification of the signaling mechanisms that control cellular responses to biochemical or mechanical stimuli upon hypertrophy. To address this challenge, Zeigler et al. developed a logic-based differential equation model of the cardiac fibroblast signaling network, which successfully validated against 80% of 41 papers from the literature not used in model development 21. This model predicted that stretch-mediated myofibroblast activation was mediated not by any single path from integrins to αSMA expression, but by an autocrine TGFβ autocrine loop. They validated this new prediction in new experiments by using a TGFβ receptor inhibitor to block cardiac myofibroblast activation in mechanically-restrained collagen gels 21. This model was later extended to predict the in vivo fibroblast dynamics after myocardial infarction, predicting how interleukin-1 can paradoxically enhance collagen production through the above autocrine TGFβ loop but suppress it through activation of NFkB and BAMBI 22. To make patient-specific predictions, Rogers et al. connected the logic-based model to transcriptional responses from valvular interstitial cells treated with patient serum samples 23. They found that endothelin-1, interleukin-6, and TGFβ were most important for explaining patient-specific fibroblast activation.

To predict therapeutic approaches, the fibroblast network model was integrated with DrugBank to predict FDA-approved drugs that could be repurposed against cardiac fibrosis 24. Interestingly, the combination drug Entresto (valsartan/sacubitril) was predicted to be particularly effective due to combined suppression of ERK through valsartan and enhancement of PKG through sacubitril 24. This prediction was validated by independent studies showing that Entresto decreases fibrosis due to pressure overload in rats 25,26 and heart failure in humans 27. Watts et al. extended this model with estrogen signaling, predicting that the effects of some drugs may be sex specific 28, building on previous experimental studies of how estrogen affects cardiac fibroblast signaling and activation 29. Others have combined network modeling with an experimental drug screen through machine learning to identify pathways by which drugs regulate new fibroblast phenotypes such as stress fiber organization 30.

While fibroblasts play a central role, they regulate fibrosis through communication with many other cell types. Jin et al. developed a mathematical model that simulated the communication between monocytes, macrophages, and fibroblasts that lead to fibrosis after myocardial infarction 31. Their model was validated against dynamics of inflammation and collagen content after myocardial infarction and then applied to predict how the strength or timing of perturbations to TGFβ or MMP9 can modulate the kinetics of post-MI fibrosis. Using a similar paradigm, Chowkwale et al. developed a model of the post-MI communication between neutrophils, monocytes, macrophages, cardiomyocytes, and fibroblasts 32. They validated the model against 61 of 84 experiments not used to build the model. Using this model, they identified key dynamic features that control inflammation, fibrosis, and a new concept of inflammation-fibrosis coupling. Specifically, they predicted that inflammation is amplified by positive feedback between neutrophils and IL1β, macrophage phagocytosis of cardiomyocytes is critical for inflammation to drive fibrosis, and that fibroblast proliferation acts as an ultrasensitive switch to amplify collagen deposition 32. Intriguingly, this dual-amplification control system identified for inflammation-fibrosis coupling 32 appears analogous to that in excitation-contraction coupling 33. Intercellular models illustrate complex dynamic relationships that should be experimentally validated for potential as therapeutic strategies.

Reviewer 3 Report

The present manuscript highlights on physiological and pathological cardiac hypertrophy in details. It also focus on the interplay of microenvironmental factors involved in the development of cardiac hypertrophy including cardiomyocytes and non-myocardial cells. The manuscript also reviewed the involvement of numerous pro-hypertrophic, pro-fibrotic, and pro-inflammatory mediators involved in cardiac hypertrophy. The understanding of all the factors associated with pathological cardiac hypertrophy will pave the way for research on treatment/preventive aspects. manuscript is well written. I found two corrections. 

1. Line 382: Change font of  Reactive oxygen species in to italics. 

2. Line 856: replace conflict of interest with abbreviations. 

Author Response

The present manuscript highlights on physiological and pathological cardiac hypertrophy in details. It also focus on the interplay of microenvironmental factors involved in the development of cardiac hypertrophy including cardiomyocytes and non-myocardial cells. The manuscript also reviewed the involvement of numerous pro-hypertrophic, pro-fibrotic, and pro-inflammatory mediators involved in cardiac hypertrophy. The understanding of all the factors associated with pathological cardiac hypertrophy will pave the way for research on treatment/preventive aspects. manuscript is well written. I found two corrections.

Response: My coworkers and I thank you for reviewing our manuscript and for your valuable comments.

1.Line 382: Change font of Reactive oxygen species in to italics.

Response: Corrected.

2. Line 856: replace conflict of interest with abbreviations.

Response: Corrected.